# A feature transferring workflow between data-poor compounds in various tasks

Xiaofei Sun[1,2], Jingyuan Zhu[4], Bin Chen[2,3]*, Hengzhi You[4]*, Huiqing Xu[5]

**1** Chengdu Institute of Computer Application, Chinese Academy of Sciences, Chengdu, Sichuan, China, **2** University of Chinese Academy of Sciences, Beijing, China, **3** IRIAI, Harbin Institute of Technology, Shenzhen, Guangdong, China, **4** School of science, Harbin Institute of Technology, Shenzhen, Guangdong, China, **5** Guangdong Energy Group Science and Technology Research Institute Co., Ltd., Guangzhou, Guangdong, China

* chenbin2020@hit.edu.cn (BC); youhengzhi@hit.edu.cn (HY)

## Abstract

Compound screening by in silico approaches has advantages in identifying high-activity leading compounds and can predict the safety of the drug. A key challenge is that the number of observations of drug activity and toxicity accumulation varies by target in different datasets, some of which are more understudied than others. Owing to an overall insufficiency and imbalance of drug data, it is hard to accurately predict drug activity and toxicity of multiple tasks by the existing models. To solve this problem, this paper proposed a two-stage transfer learning workflow to develop a novel prediction model, which can accurately predict drug activity and toxicity of the targets with insufficient observations. We built a balanced dataset based on the Tox21 dataset and developed a drug activity and toxicity prediction model based on Siamese networks and graph convolution to produce multitasking output. We also took advantage of transfer learning from data-rich targets to data-poor targets. We showed greater accuracy in predicting the activity and toxicity of compounds to targets with rich data and poor data. In Tox21, a relatively rich dataset, the prediction model accuracy for classification tasks was 0.877 AUROC. In the other five unbalanced datasets, we also found that transfer learning strategies brought the accuracy of models to a higher level in understudied targets. Our models can overcome the imbalance in target data and predict the compound activity and toxicity of understudied targets to help prioritize upcoming biological experiments.

## Introduction

Determining the intricate meanings of the information in chemical molecular systems from chemical structures is important for finding chemicals with favorable pharmacological, toxicological, and pharmacokinetic properties [1–5]. Existing studies show that screening potential drugs by predicting activity and toxicity in leading compounds can be of great help [6–11]. A considerable number of drug activity prediction methods based on machine learning have been investigated, including naive Bayes [6, 12, 13], logistic regression [7, 8], k-nearest

**Data Availability Statement:** All datasets used in this study (Tox21, Toxcast, PCBA, MUV, HIV, and Freesolv) are publicly available online.

**Funding:** This work is partially supported by the Shenzhen Science and Technology Research Fund

(JCYJ20190806142203709; JSGG20191129114029286; JSGG20201103153807021), Talent Development Starting Fund from Shenzhen Government (HA11409030), and Guangdong Province Basic and Applied Basic Research Fund Project (2021A1515110366). There was no additional external funding received for this study. The funder provided support in the form of salaries for authors [XS, JZ, BC, HY and HX], but did not have any additional role in the study design, data collection and analysis, decision to publish, or preparation of the manuscript. The specific roles of these authors are articulated in the 'author contributions' section.

**Competing interests:** The authors have declared that no competing interests exist.

neighbors [9], support vector machines [10, 11, 14–16], random forests [11, 17, 18], and artificial neural networks [16]. These methods have contributed significantly to the development of drug activity and toxicity prediction, but the problem of scarcity and imbalance of target data has not received sufficient attention. Recently, the advent of deep learning approaches has shown a significant impact on this traditional cheminformatics task due to their enormous capacity to learn the structure and properties of compounds [19–27]. These studies used descriptor-based or graph-based methods to predict the activity and toxicity of compounds to targets on publicly available datasets. The emergence of these latest approaches has further enhanced the effectiveness of drug activity and toxicity prediction, but the scarcity and imbalance of target data remain a challenge.

A key challenge in the development of generalizable drug virtual screening models is the imbalance in target/task data, wherein the accumulated number of drug activities varies from target to target and the number of positive drugs is very rare [28]. In the Tox21 dataset [29], for example, targets such as GPCRs (G protein-coupled receptors), nuclear receptors, ion channels, and kinases have rich data on drug toxicity, while other targets have less data. Surprisingly, the imbalance in target data is more pronounced in many other datasets (Fig 1). Existing methods use original target and drug data without balancing or augmentation due to an 'activity cliff', which is a pair of compounds with high structural similarity but unexpectedly high activity differences [6–26]. In this work, we attempt to address the challenge of data imbalance and data scarcity to accurately predict drug activity and toxicity.

There are a large number of understudied targets in clinical and scientific research (Fig 1), such as ADRB2, OPRK1, and PPARRG. Many obstacles hinder drug activity and toxicity studies at these targets. For example, due to the difficulty of obtaining tumor tissue fragments of bone metastasis, the scarcity of sarcomatous-type tumors, and the difficulty in culturing bone tissue into cell lines, the number of cell models is insufficient [30, 31]. Due to the shortage of these cell line models, corresponding high-throughput screening is difficult to carry out, and target studies

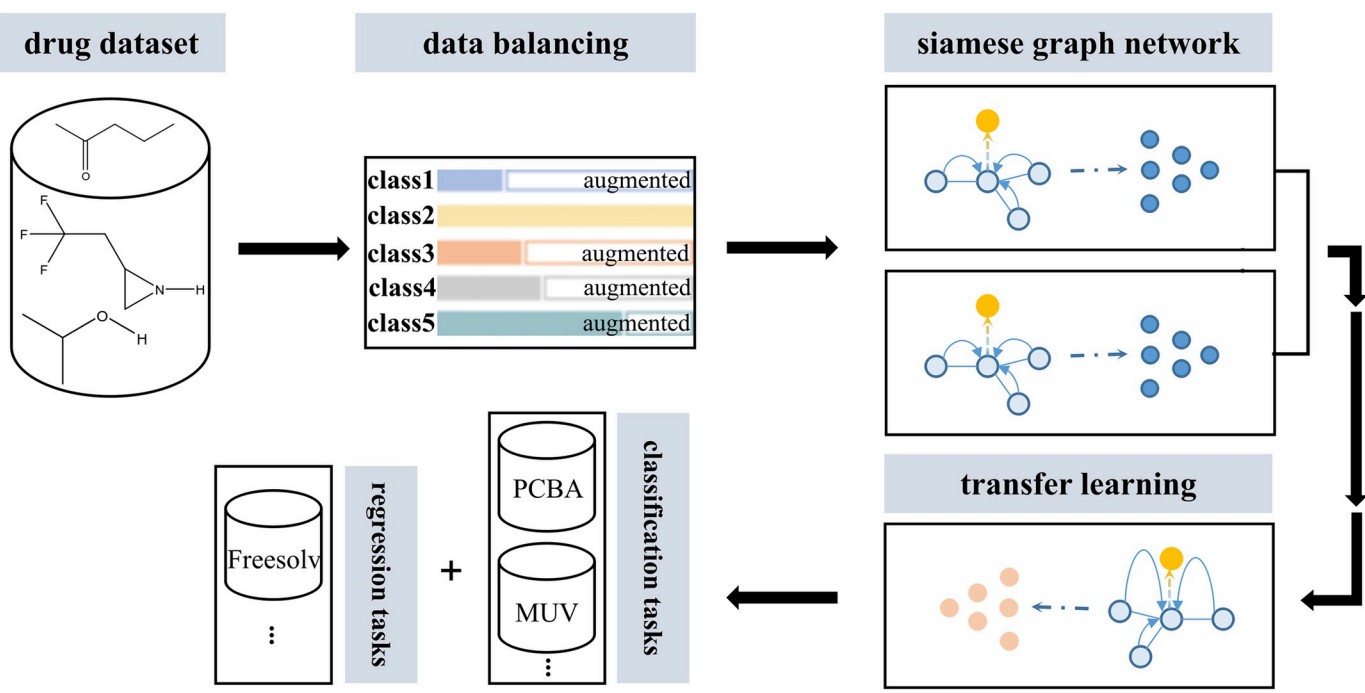

**Fig 1. Summary of the Siamese graph convolutional network-based transfer learning workflow (SGT).**

are also greatly restricted. Therefore, there is an urgent need to develop a generalizable drug activity and toxicity prediction tool to promote the understanding of understudied targets.

We are trying to develop a generalizable drug activity and toxicity prediction model to address the challenge of data scarcity for understudied targets. The training data for understudied targets were insufficient: in addition to the small total number of their experimental observations, the positive rate of drug activity was also very low. The potential solution to this data scarcity problem is to take advantage of information from data-rich targets to data-poor targets. Because these different targets have biological commonality, the drugs have similar activity and toxicity to them to some extent [31–33]. Therefore, we proposed utilizing the drug activity and toxicity of data-rich targets through transfer learning to help improve the performance of the model on data-poor targets. Transfer learning is the recognition of knowledge and skills learned from previous domains/tasks and their application to new domains/tasks. To learn about the generalizable expression of drug activity and toxicity to targets, we selected Tox21, which is relatively data-rich (high number of experimental observations and high drug-positive rates), to produce a balanced dataset. We used the drugs and targets in our balanced Tox21 dataset to train the Siamese graph convolutional neural network model and verified our model in five datasets: ToxCast, HIV, MUV, PCBA, and FreeSolv after transfer learning (Fig 1).

## Materials

### A balanced dataset

Our balanced dataset is based on the Tox21 dataset, which is designed to help scientists understand the potential of the chemicals and compounds being tested through the Toxicology in the 21st Century initiative to disrupt biological pathways in ways that may result in toxic effects [2, 7–9, 14, 19–24, 29]. The numbers of targets and unique compounds were 12 and 7831, respectively. There were a total of 93972 compound toxicity experimental observations (i.e., a pair of compounds and targets), of which the number of drugs and the positive rate of drugs varied from target to target. For a target in the Tox21 dataset, we first took a pair of positive compounds, then a pair of negative compounds, and then a positive compound and a negative compound. We repeatedly obtained pairs of data until a new balanced dataset for the Siamese network was built. The dataset contains 23,493 toxicity experimental data. We compared the model using this balanced dataset with the baseline study using Tox21 [34].

We used five other datasets for verification (Fig 2): (i) ToxCast: another dataset from the same initiative as Tox21 that uses virtual high-throughput screening to provide toxicology data. It contains qualitative results from more than 600 experiments on 8615 compounds [29]. (ii) MUV: a dataset designed for virtual screening technology that contains approximately 90,000 compounds involving 17 tasks, and the positive compounds in this dataset are structurally distinct from each other [34, 35]. (iii) PCBA: PubChem BioAssay [35–38] is a dataset of small-molecule biological activity produced by high-throughput screening. We used a subset of PubChem in MoleculeNet containing 400,000 compounds and 128 biological assays. (iv) HIV: The HIV dataset [34, 37], an AIDS antiviral screening dataset introduced by the Drug Treatment Program (DTP), tested the ability of 41,913 compounds to suppress HIV. (v) FreeSolv: The Free Solvation Database [35], used in the SAMCL Blind Prediction Challenge, contains the hydration free energy of 643 compound molecules.

### Compound features

SMILES is a linear representation of molecular structures that uniquely describes a molecule with an ASCII string. SMILES uses atomic symbols to represent the atoms themselves and

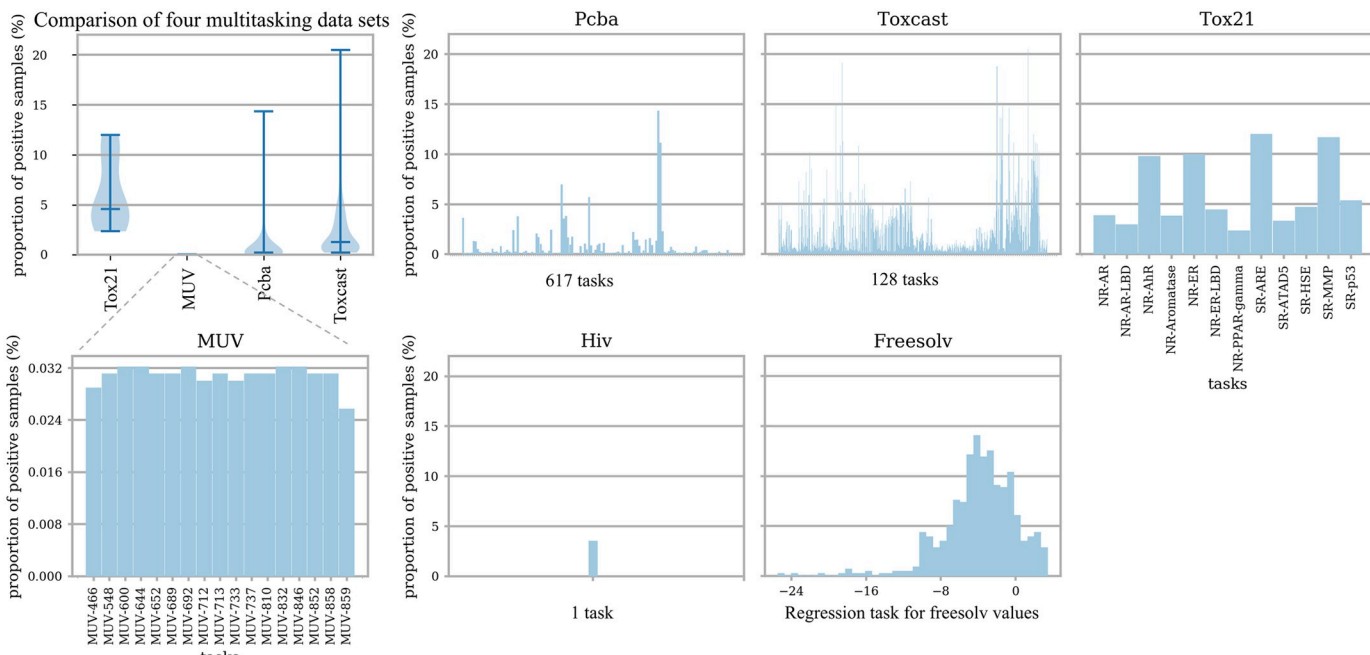

**Fig 2. Data distribution of the drug datasets.** The first graph in the first row is an overview of the proportions of positive drug samples of the targets in datasets a Tox21, b MUV, c PCBA, and d Toxcast, and other graphs show in detail the distribution of positive samples in each dataset.

special characters to represent the relationships between atoms. We used Rdkit (http://www.rdkit.org) to process SMILES into rich atomic features such as degree, implicit valence, formal charge, number of radical electrons, and adjacency list and then input these features into the compound encoder. The molecular features of the compound were extracted by the compound encoder detailed in a later section.

## Methods

### Method overview

Our transfer learning method for enhancing drug activity and toxicity prediction with scarce data is to transfer parameters of models pretrained using a balanced training set to a prediction model for the specific targets. We first trained the toxicity prediction model of data-rich targets from a balanced dataset containing the drug toxicity of 12 different targets. The Siamese network facilitates the integration of two network structures in a parameter-sharing manner, benefiting the learning of data-poor targets, so we used the Siamese network in pretraining the model. In addition, the graph convolutional neural network has obvious advantages in obtaining molecular representation, which we took as a feature extractor in the Siamese network. We initialized the prediction model of data-poor targets using the parameters of the pretrained model and retrained it with the corresponding data.

### Method overview

Our Siamese graph convolutional neural network (Fig 3a) determines whether a pair of compounds belongs to the same class by the distance between them and produces a corresponding probability distribution through the decision layer. We formulated this prediction problem. Given an input block $x := \{C_i, C_J\}$ of compound $C_i$ and compound $C_j$, the prediction model $f$ is

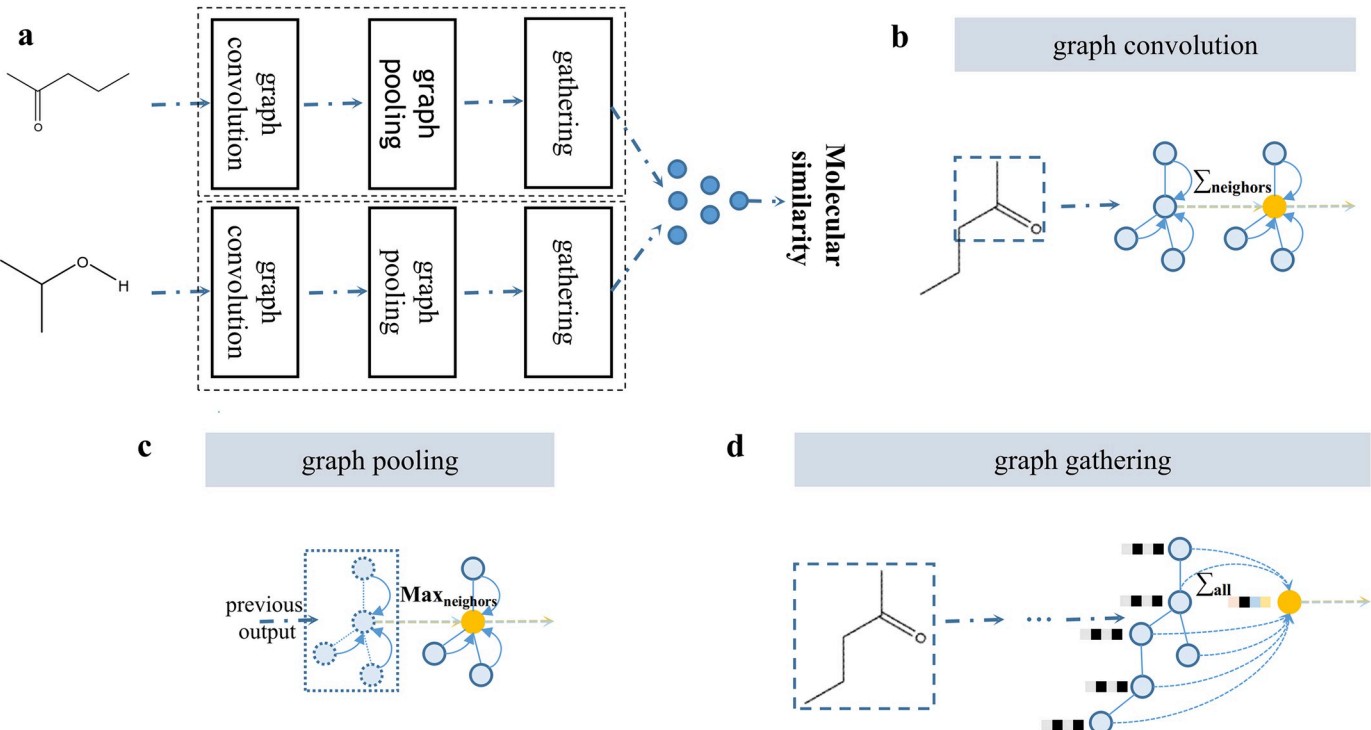

**Fig 3. A graphical representation of the network described in this article. a** Siamese graph convolutional neural network with shared weights, **b** graph convolution operation, **c** graph pooling operation, and **d** graph gathering operation in the network.

a function such that

$$y_{ij} \approx f(C_i, C_j) \tag{1}$$

where $y_{ij}$ is the similarity score of two compounds. We used two compound encoders that share parameters to form a Siamese network and trained it on a balanced dataset to generate a pretrained Siamese graph model that can be used for subsequent transfer learning.

## Compound encoder

We used graph convolutional networks (Fig 3b) to encode SMILES sequences because the latest studies [35–38] suggest that graph convolutional networks have an advantage in processing molecular structures. The compound encoder learns the molecular representation of each compound. Considering that the number of atoms in a molecule changes widely and the value of degree is relatively limited, the compound encoder stores and calculates atoms by degree. Each atomic feature, such as implicit valence, is first represented as a one-hot vector, and then the one-hot vectors of all atomic features are combined and fed into the compound encoder.

A compound could be considered a graph, its nodes representing atoms, and the edges that connect them together are bonds. In graph convolution [36], for a node, we feed its features and neighbors into two dense layers and then add the output of the dense layers as the new features of the node. The calculations of the new features of nodes with the same degree share weights. In a compound, if an atom $a$ has a total of $n$ neighbors, its new features after graph

convolution can be formulated as

$$a' = \sigma\left(W_a a + \sum_{j=1}^{n} W_r r_i + b\right) \tag{2}$$

where $W_a$ is the weight of node $a$; $W_r$ is the weight of the neighboring nodes; $b$ is bias, and $\sigma$ is the activation function ReLU. The yellow arrows represent the dense layers of atomic $a$ and its neighbors, with weights of $W_a$ and $W_r$, respectively.

Similar to the convolutional neural network, the graph pooling layer (Fig 3c) is used in the compound encoder. Graph pooling is the operation of returning the largest or average feature among an atom and its neighbors. Graph pooling increases the receptive field without adding additional parameters:

$$a_{pool} = max\{a, r_i\}, i = 1, \ldots, n \tag{3}$$

In graph convolution and graph pooling, each atom has a descriptor vector. However, to make a final prediction, a fixed-size vector descriptor for the entire graph will be required. The graph gather layer (Fig 3d) sums all the feature vectors of all atoms in the compound molecule to obtain the molecular feature vector:

$$m_{gather} = \sum_{a \in A} a \tag{4}$$

## Tox21 prediction model

Our goal is to predict the activity and toxicity of unobserved compounds at a given target and to prioritize compounds that can be used in experiments. To achieve this, we used transfer learning in addition to Siamese graph convolutional neural networks and data balancing. We transferred the parameters of the drug encoder in the Siamese graph model to a new prediction model, which we call the Tox21 prediction model, and retrained it using the Tox21 dataset. Our Tox21 prediction model consists of a drug encoder and a decision network (Fig 4) for toxicity prediction in an end-to-end manner.

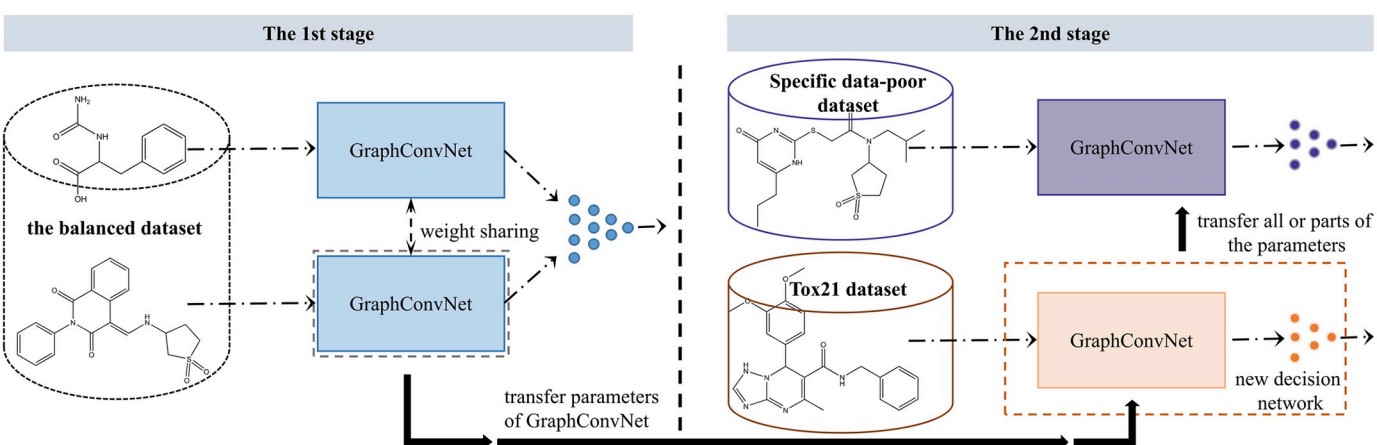

**Fig 4. The transfer learning workflow for data-poor targets.** Two stages are used to transfer model parameters from data-rich targets to data-poor targets in specific datasets.

### Fine-tuning to data-poor targets

We focused on transfer learning from data-rich targets in a balanced dataset to targets whose data are scarce in other unbalanced datasets (Fig 4). In the transfer learning experiments, we selected the ToxCast, HIV, MUV, PCBA, and FreeSolv datasets (Fig 2) because most of their targets had fewer observations and low positive rates, but the total number of observations was sufficient to build a prediction model for comparative experiments. The drug toxicity prediction model in the first stage, which is pretrained with data-rich Tox21 targets, learns the underlying mechanism between compounds and targets, and then the prediction model for specific data-insufficient targets transfers the pretrained model parameters from the Tox21 model and fine-tunes the parameters. Optional fine-tuning strategies include **(i)** retraining all parameters, **(ii)** fixing the compound encoder, retraining the decision layer, and **(iii)** not retraining at all. By comparing the performance of these three different strategies in unbalanced datasets, the first strategy was found to be optimal, so we used the first fine-tuning strategy in subsequent experiments.

### Training loss

Our Siamese graph convolutional neural network learns multiple tasks on a balanced dataset, and the training loss is the contrastive loss:

$$L_w(Y, \overrightarrow{X_1}, \overrightarrow{X_2}) = (1 - Y)(D_w)^2 + Y\{max(0, m - D_w)\}^2 \tag{5}$$

where $Y$ is the label of whether two compounds are the same class; $m$ is the threshold, and $D_w$ represents the Euclidean distance of the two compound features $\overrightarrow{X_1}$ and $\overrightarrow{X_2}$:

$$D_w(\overrightarrow{X_1}, \overrightarrow{X_2}) = \|f(\overrightarrow{X_1}) - f(\overrightarrow{X_2})\|_2 \tag{6}$$

The training loss for the Tox21 model and fine-tuning were similarly defined:

$$L = -\sum_{j=1}^{T} y_j log S_j \tag{7}$$

where $S_j$ is a softmax function:

$$S_j = \frac{e^{a_j}}{\sum_{k=1}^{T} e^{a_k}} \tag{8}$$

We first minimized $L_w$ loss with all training batches in the balanced dataset and then switched to $L$ in fine-tuning for training. Optimizer was Adam in TensorFlow 1.6.0.

### Evaluation

#### Training and test set

We evaluated the prediction models with external validation on unbalanced datasets with data-poor targets. For datasets with scarce target data such as ToxCast, HIV, MUV, PCBA, and FreeSolv, we set aside some of the data as independent test sets (20%) and the rest as training sets (80%). The drugs in the test set do not overlap with the training set. Each data-poor target had a small number of distinct drugs, and different targets sometimes shared drugs. Note that each training or test set of the Siamese graph convolutional network was a tuple {$d_i$, $d_j$} of drug pairs for certain targets, while each training or test set in fine-tuning was a drug.

## Accuracy measures

We used different criteria to measure the accuracy of regression and classification tasks. The classification accuracy measure was the area under the ROC curve (AUC); the regression accuracy measures were $R^2$, RMSE, and MAE.

## Baselines

We compared the accuracy of our classification tasks with five baseline methods: **(i)** Attentive FP, which is a new graph neural network architecture for molecular representation and uses graph attention mechanisms to learn from a drug discovery dataset [37]. **(ii)** Neural FP is a kind of circular fingerprint that can be generated differentiably by a neural network, which allows end-to-end learning of molecules of any size and shape [37]. **(iii)** GraphConv directly uses the molecular connectivity graph as input and provides a learnable featurization process that extracts meaningful representations of molecules [34, 39]. **(iv)** ECFP+LR that utilizes linear regression using molecular extended-connectivity fingerprints (ECFPs) on specific tasks [34]. **(v)** SVM that maps input vectors to a high-dimensional feature space through nonlinear mapping, where an optimal separating hyperplane is constructed to separate the samples [34]. The benchmark accuracy values of various methods (including Attentive FP, Neural FP, GraphConv, ECFP-LR, and SVM) are listed in the referenced papers [34, 37].

## Results

### Prediction accuracy of data-rich targets

We used three split methods, index, random, and scaffold, to split the dataset into a training set, validation set, and test set. We first evaluated the accuracy of the model trained and tested using data-rich targets in Tox21 (Fig 5a). We achieved better performance on these datasets than the baseline approaches. On the test sets generated using these three split methods, we achieved AUROCs of 0.833, 0.877, and 0.761, while Neural FP, Attentive FP, and ECFP-LR achieved AUROCs of 0.829, 0.858, and 0.755 on the test set generated using the random method. We found that our Siamese graph convolutional neural network has significant AUROC improvements over GraphConv in the training, validation, and test sets. This proves the advantages of data balancing and Siamese graph convolutional neural networks.

### Prediction accuracy of a specific model with transfer learning

We then experimented with poor data targets in the FreeSolv (Fig 5b), MUV, PCBA (Fig 6), and ToxCast, HIV (Fig 7) datasets. We found that proper transfer learning can improve the accuracy of drug activity and toxicity prediction at targets with poor data. In the classification tasks of the PCBA, MUV, ToxCast, and HIV datasets, we achieved AUROCs of 0.858, 0.715, 0.655, 0.804, 0.851, 0.747, 0.725, 0.715, 0.655, and 0.772, 0.856, 0.805 using the index, random, and scaffold split methods, respectively. In the regression task of FreeSolv datasets, we achieved 0.953 R2, 0.778 RMSE, 0.539 MAE, 0.936 R2, 0.899 RMSE, 0.585 MAE, and 0.755 R2, 1.85 RMSE, 1.519 MAE, respectively (Figs 5b and 8). This method can also make the prediction of each task consistent with the observed value, and improve the unbalanced performance of the model between tasks (Figs 9–11). The area under the ROC curve (AUC) of various models in the Tox21, ToxCast, MUV, and HIV datasets is shown in Table 1. In addition to R2, RMSE and MAE (kcal/mol) are also provided to perfectly reflect the performance of the models in FreeSolv data set (Table 2).

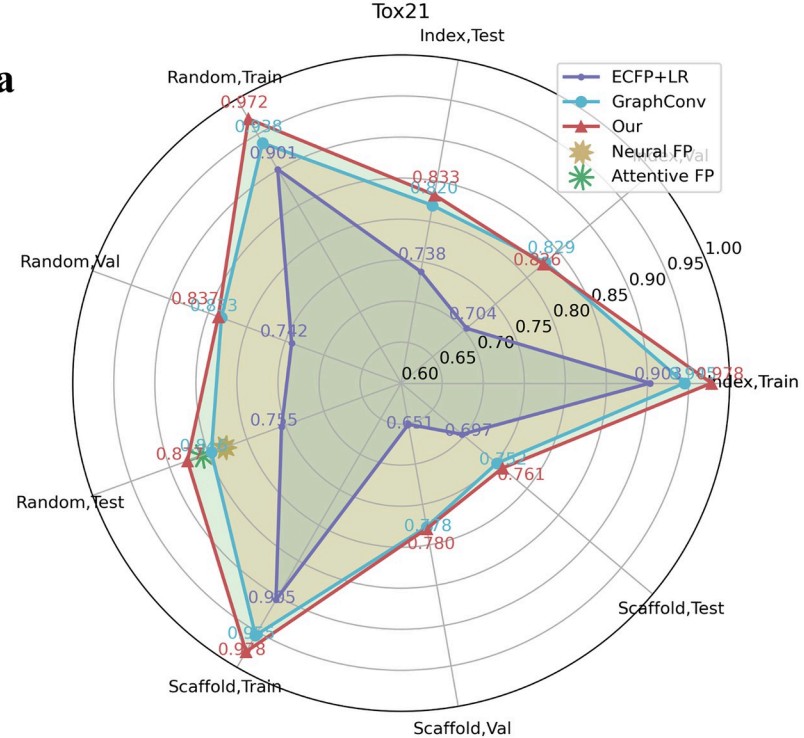

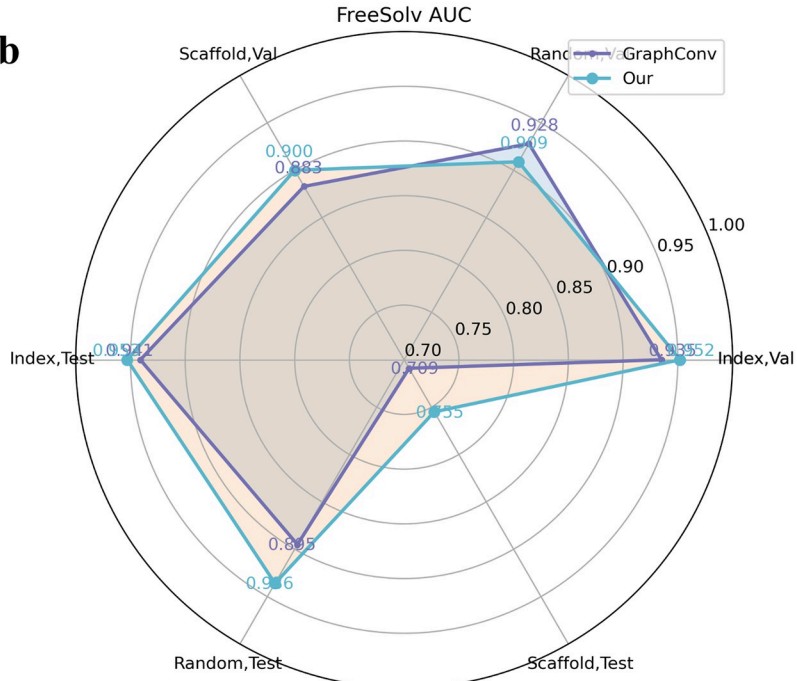

**Fig 5. Performance comparison between our model and the baseline model in multitask classification and regression tasks.** The area under the curve (AUC) of the ROC curve of various models in the **a** Tox21 and **b** Freesolv datasets.

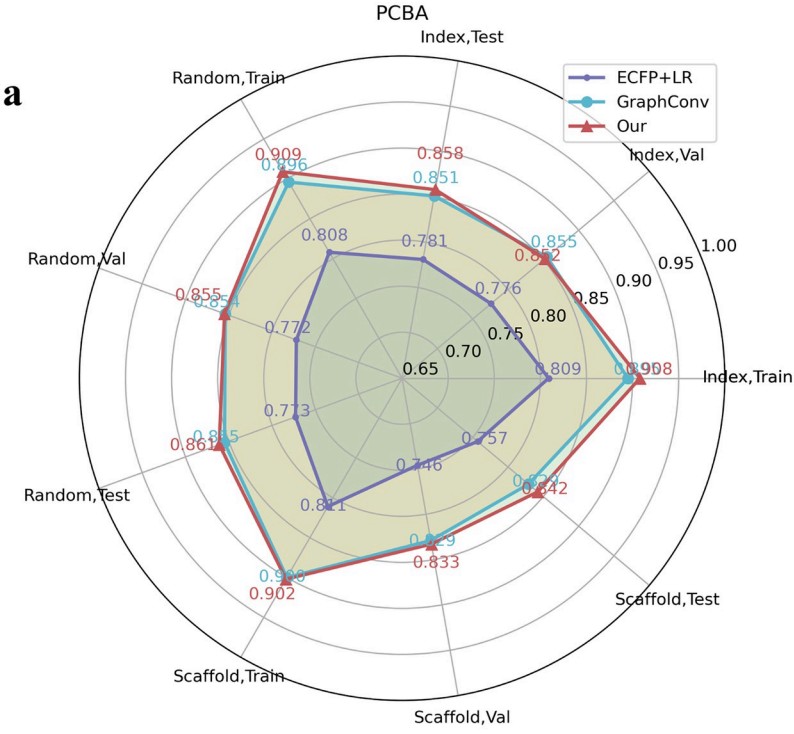

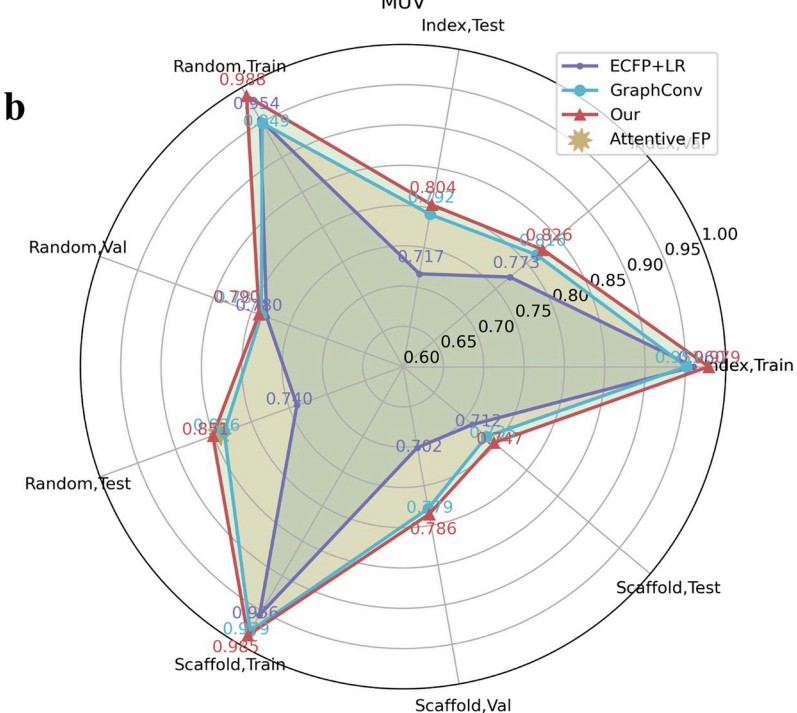

**Fig 6. Performance comparison between our model and the baseline model in multitask classification tasks.** The area under the curve (AUC) of the ROC curve of various models in the **a** PCBA and **b** MUV datasets.

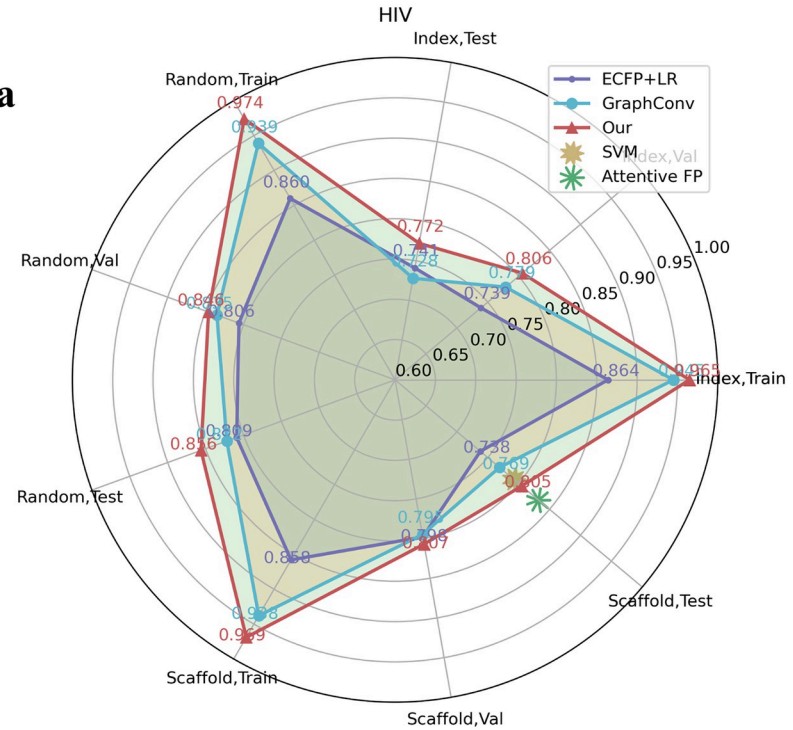

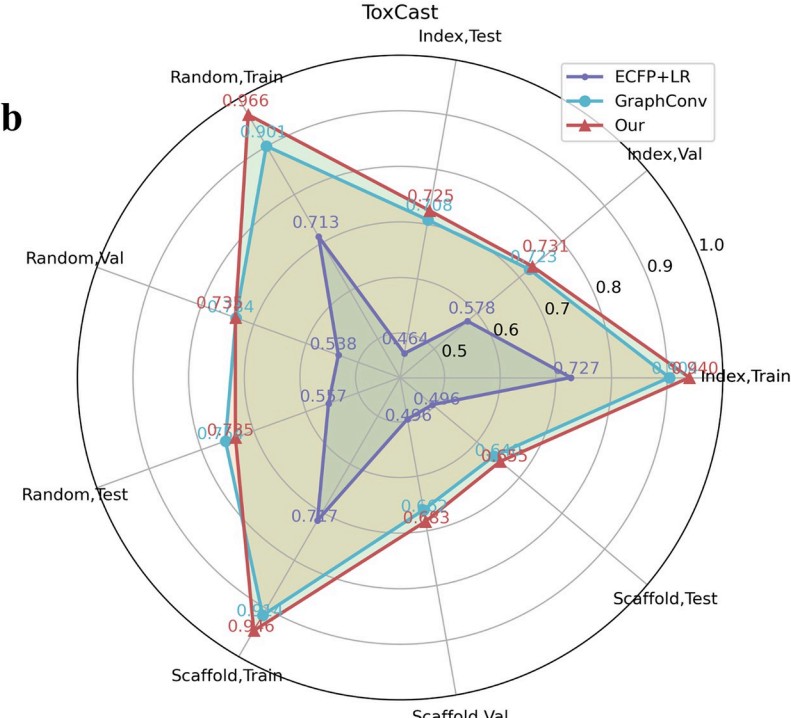

**Fig 7. Performance comparison between our model and the baseline model in multitask classification tasks.** The area under the curve (AUC) of the ROC curve of various models in the **a** HIV and **b** Toxcast datasets.

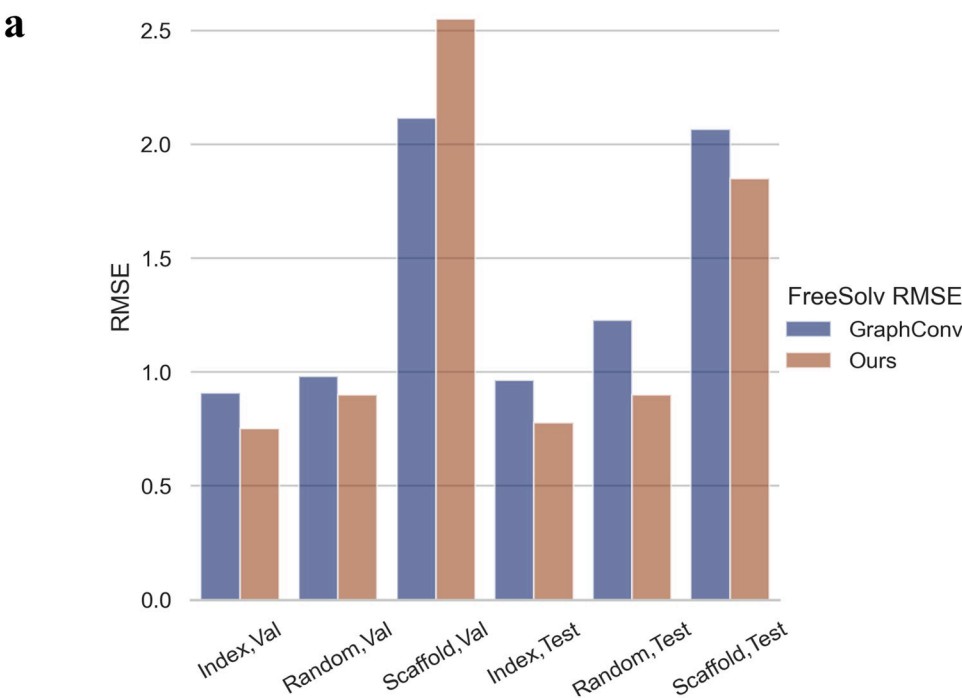

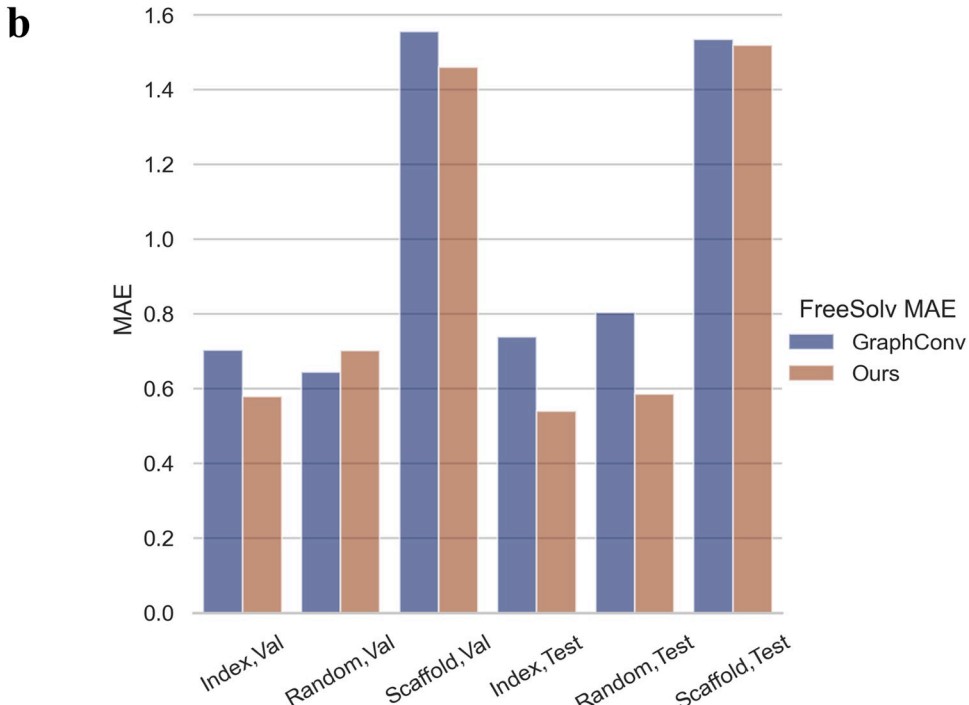

**Fig 8. Performance comparison between our model and the baseline model in the regression task. a** RMSE and **b** MAE (kcal/mol) are provided to perfectly reflect the performance of the models in the regression task of the Freesolv dataset.

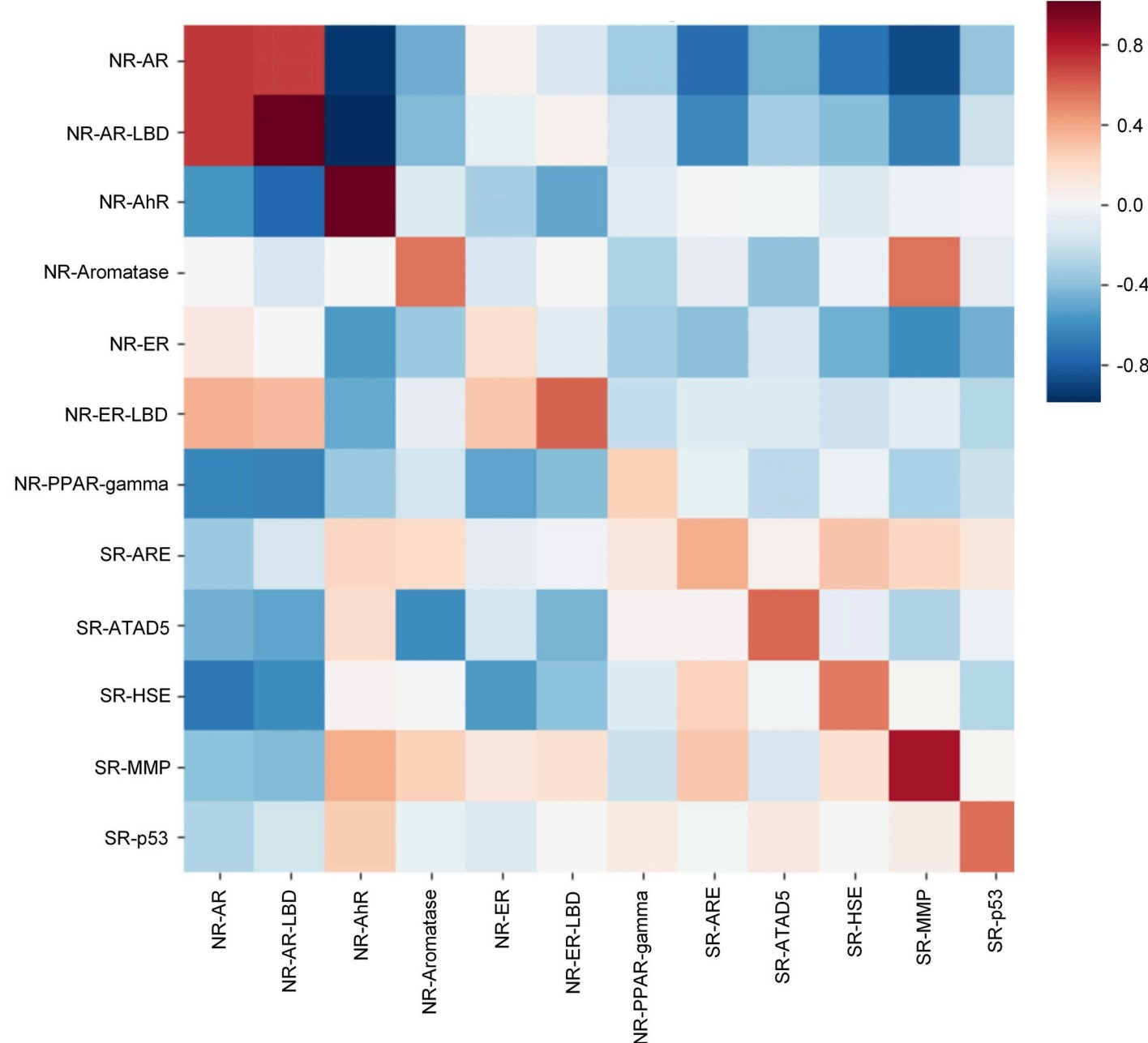

**Fig 9. The correlation between predictions and observations of our model on different tasks in tox21 data set (index split).**

## Discussion

The purpose of this study was to develop drug activity and toxicity prediction models that can even be used for targets with poor data. To this end, we **(i)** created a balanced dataset from the Tox21 dataset, which is relatively data-rich for the targets, **(ii)** integrated the graph convolution and Siamese network and used it to train a model of the toxicity of compounds to targets using the physical and chemical features of compounds in the balanced dataset, and **(iii)** transferred the toxicity prediction model from data-rich targets to data-poor targets. Finally, the

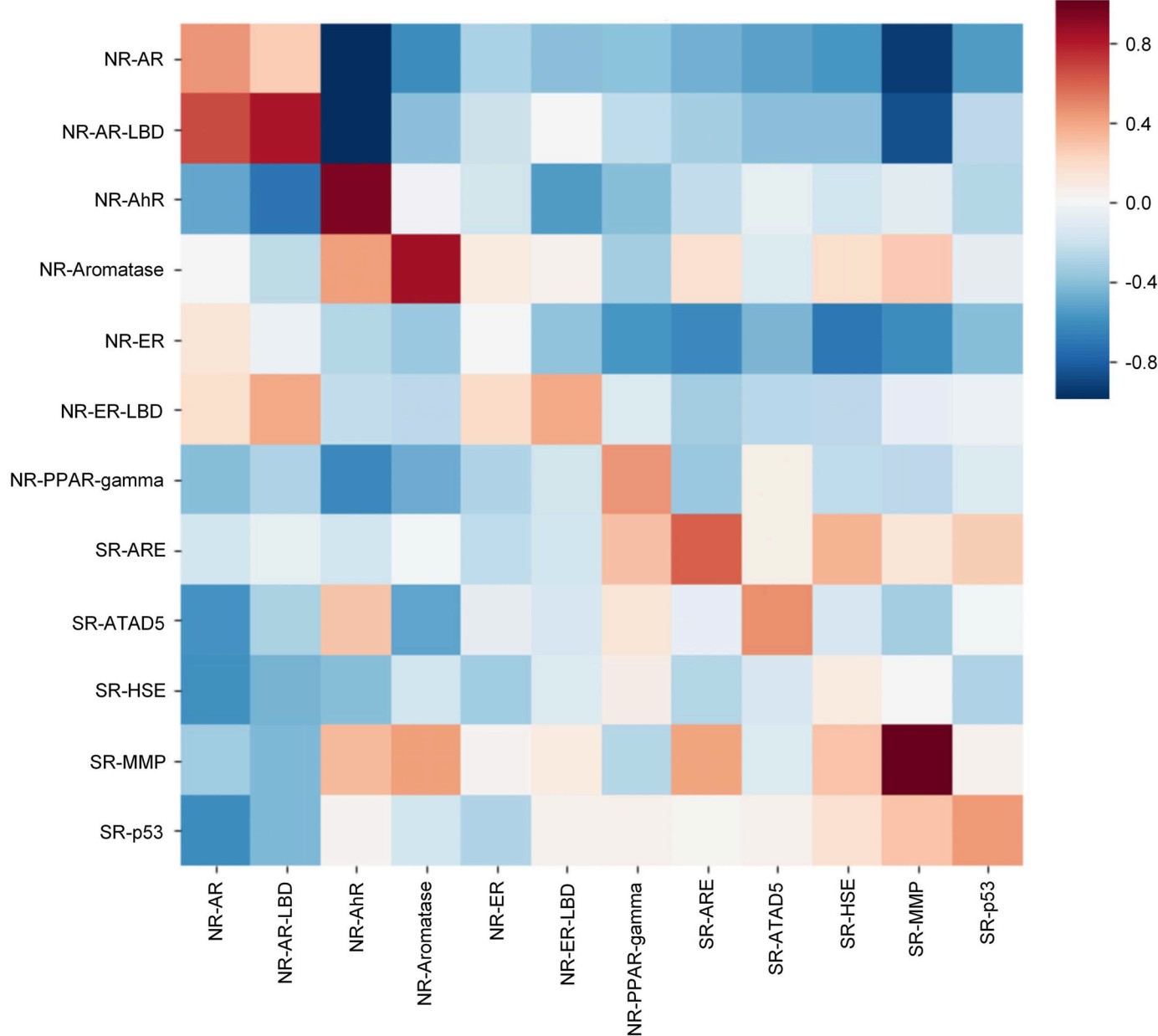

**Fig 10. The correlation between predictions and observations of our model on different tasks in tox21 data set (random split).**

proposed models more accurately predicted the activity and toxicity of compounds in the other five datasets than existing methods using an appropriate transfer learning strategy.

Our main contribution is that we approached understudied targets for drug activity and toxicity prediction. Balanced target data are the strongest support for estimating drug activity and toxicity but are only available when sufficient study is done. Inadequate research on these targets leads to an imbalance in the corresponding drug activity and toxicity data, which consequently becomes an obstacle to the development of drugs for these targets. Our drug activity and toxicity prediction models successfully address the lack and imbalance of target data while achieving competitive accuracy.

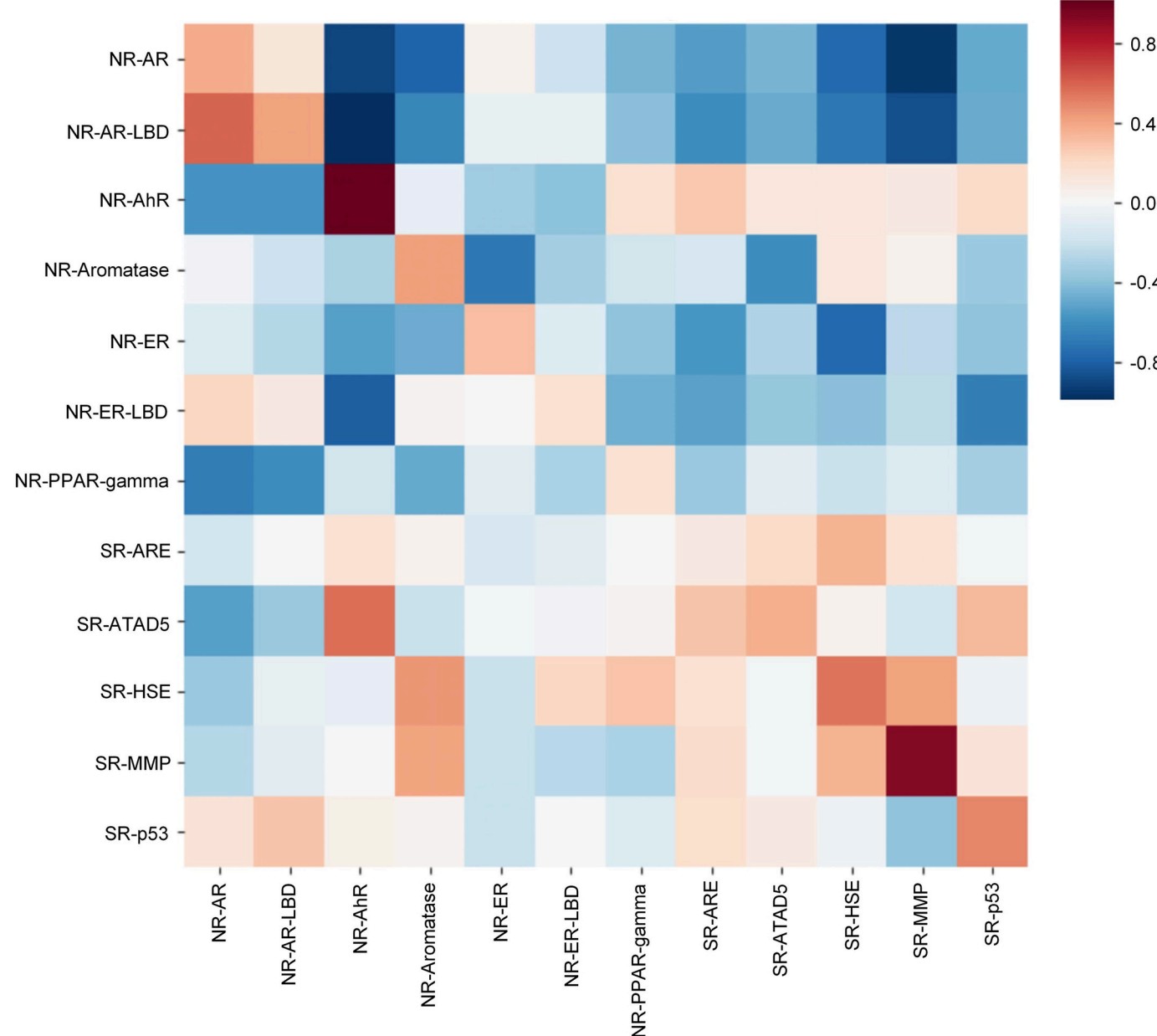

**Fig 11. The correlation between predictions and observations of our model on different tasks in tox21 data set (scaffold split).**

Although our study focused on predicting the activity and toxicity of understudied targets, our models showed greater accuracy than baseline models, even in general data-rich targets. This improved accuracy is due to the use of data balancing and the Siamese graph network in the first stage of our transfer learning. We created a balanced dataset from the Tox21 dataset, which is relatively rich in target data, for a pretrained model that combines graph convolution and the Siamese network. Balanced data allow us to maximize the few-shot learning capabilities of Siamese graph networks. In addition, this model has a good ability to indicate the correlation between molecular substructure and toxicity or activity (Fig 12).

**Table 1. The area under curve (AUC) of the ROC curve of various models in Tox21, ToxCast, MUV and HIV data sets.**

| Data Set | Model | Split Method | Train | Valid | Test |
|---|---|---|---|---|---|
| Tox21 | ECFP+LR | Index | 0.903 | 0.704 | 0.738 |
| | | Random | 0.901 | 0.742 | 0.755 |
| | | Scaffold | 0.905 | 0.651 | 0.697 |
| | GraphConv | Index | 0.945 | 0.829 | 0.820 |
| | | Random | 0.938 | 0.833 | 0.846 |
| | | Scaffold | 0.955 | 0.778 | 0.752 |
| | Neural FP | Random | - | - | 0.829 |
| | Attentive FP | Random | - | - | 0.858 |
| | Our (general model) | Index | 0.978 | 0.826 | 0.833 |
| | | Random | 0.972 | 0.837 | 0.877 |
| | | Scaffold | 0.978 | 0.780 | 0.761 |
| ToxCast | ECFP+LR | Index | 0.727 | 0.578 | 0.464 |
| | | Random | 0.713 | 0.538 | 0.557 |
| | | Scaffold | 0.717 | 0.496 | 0.496 |
| | GraphConv | Index | 0.904 | 0.723 | 0.708 |
| | | Random | 0.901 | 0.734 | 0.754 |
| | | Scaffold | 0.914 | 0.662 | 0.640 |
| | Our (Transfer learning) | Index | 0.940 | 0.731 | 0.725 |
| | | Random | 0.966 | 0.735 | 0.715 |
| | | Scaffold | 0.946 | 0.683 | 0.655 |
| PCBA | ECFP+LR | Index | 0.809 | 0.776 | 0.781 |
| | | Random | 0.808 | 0.772 | 0.773 |
| | | Scaffold | 0.811 | 0.746 | 0.757 |
| | GraphConv | Index | 0.895 | 0.855 | 0.851 |
| | | Random | 0.896 | 0.854 | 0.855 |
| | | Scaffold | 0.900 | 0.829 | 0.829 |
| | Our (Transfer learning) | Index | 0.908 | 0.852 | 0.858 |
| | | Random | 0.909 | 0.855 | 0.861 |
| | | Scaffold | 0.902 | 0.833 | 0.842 |
| MUV | ECFP+LR | Index | 0.960 | 0.773 | 0.717 |
| | | Random | 0.954 | 0.780 | 0.740 |
| | | Scaffold | 0.956 | 0.702 | 0.712 |
| | GraphConv | Index | 0.951 | 0.816 | 0.792 |
| | | Random | 0.949 | 0.787 | 0.836 |
| | | Scaffold | 0.979 | 0.779 | 0.735 |
| | Attentive FP | Random | - | - | 0.843 |
| | Our (Transfer learning) | Index | 0.979 | 0.826 | 0.804 |
| | | Random | 0.988 | 0.790 | 0.851 |
| | | Scaffold | 0.985 | 0.786 | 0.747 |
| HIV | ECFP+LR | Index | 0.864 | 0.739 | 0.741 |
| | | Random | 0.860 | 0.806 | 0.809 |
| | | Scaffold | 0.858 | 0.798 | 0.738 |
| | SVM | Scaffold | - | - | 0.792 |
| | GraphConv | Index | 0.945 | 0.779 | 0.728 |
| | | Random | 0.939 | 0.835 | 0.822 |
| | | Scaffold | 0.938 | 0.795 | 0.769 |
| | Attentive FP | Scaffold | - | - | 0.832 |
| | Our (Transfer learning) | Index | 0.965 | 0.806 | 0.772 |
| | | Random | 0.974 | 0.846 | 0.856 |
| | | Scaffold | 0.969 | 0.807 | 0.805 |

**Table 2. The Performances in FreeSolv data set.** $R^2$, RMSE and MAE (kcal/mol) are provided to reflect the performance of the models.

| Model | Split Method | Valid | | | Test | | |
|---|---|---|---|---|---|---|---|
| | | $R^2$ | RMSE | MAE | $R^2$ | RMSE | MAE |
| GraphConv | Index | 0.935 | 0.909 | 0.703 | 0.941 | 0.963 | 0.738 |
| | Random | 0.928 | 0.982 | 0.644 | 0.895 | 1.228 | 0.803 |
| | Scaffold | 0.883 | 2.115 | 1.555 | 0.709 | 2.067 | 1.535 |
| Our (Transfer learning) | Index | 0.952 | 0.751 | 0.579 | 0.953 | 0.778 | 0.539 |
| | Random | 0.909 | 0.900 | 0.701 | 0.936 | 0.899 | 0.585 |
| | Scaffold | 0.900 | 2.55 | 1.460 | 0.755 | 1.85 | 1.519 |

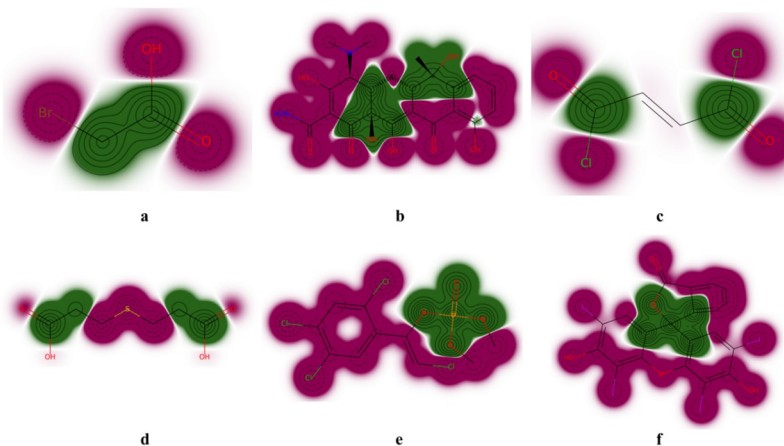

**Fig 12. An example diagram showing the correlation between atomic and molecular toxicity using similarity maps.** Red represents a higher correlation, and green represents a lower correlation.

The limitation of this study is that the model cannot achieve high accuracy (no more than 80%) in the ToxCast dataset. A possible explanation for this might be that the ToxCast dataset has the largest number of targets (617), the largest imbalance, and low drug positive rates (with positive rates ranging from 0–19% between targets), so the presence of a large number of understudied targets may result in discrepancies in model predictions and observations.

## Conclusion

In summary, our model is an end-to-end prediction model of drug activity and toxicity, learning the interaction between drugs and targets. Based on the fact that similar gene expression is shared by different target tissues and therefore drugs exhibit activity and toxicity to the targets in a similar manner, we used a siamese graph convolutional neural network and transfer learning from data-rich targets to data-poor targets to enable prediction models to play a role in data-poor targets. For future work, our drug activity and toxicity prediction models can reveal the underlying mechanisms of interaction between other potential targets and drugs and provide new methods for efficient and low-cost drug discovery. In order to make the model proposed in this paper can be used in common situation, it is essential to establish massive multi-task datasets for pretraining, which is also an interesting attempt for our future research work.

## Author Contributions

**Conceptualization:** Xiaofei Sun.

**Data curation:** Xiaofei Sun, Jingyuan Zhu.

**Formal analysis:** Xiaofei Sun, Jingyuan Zhu.

**Funding acquisition:** Hengzhi You.

**Methodology:** Xiaofei Sun.

**Project administration:** Bin Chen.

**Resources:** Bin Chen, Hengzhi You.

**Software:** Xiaofei Sun.

**Supervision:** Bin Chen, Hengzhi You, Huiqing Xu.

**Validation:** Xiaofei Sun, Jingyuan Zhu, Bin Chen, Hengzhi You.

**Visualization:** Xiaofei Sun, Huiqing Xu.

**Writing – original draft:** Xiaofei Sun.

**Writing – review & editing:** Jingyuan Zhu, Bin Chen, Hengzhi You.

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
