## [Decision Letter · Decision Letter 0]

9 Feb 2022

PONE-D-22-00703A feature transferring workflow between data-poor compounds in various tasksPLOS ONE

Dear Dr. Chen,

Thank you for submitting your manuscript to PLOS ONE. After careful consideration, we feel that it has merit but does not fully meet PLOS ONE’s publication criteria as it currently stands. Therefore, we invite you to submit a revised version of the manuscript that addresses the points raised during the review process.

We look forward to receiving your revised manuscript.

Kind regards,

Sathishkumar V E

Academic Editor

PLOS ONE

Journal Requirements:

"This work is partially supported by Shenzhen Science and Technology Research Fund (JCYJ20190806142203709; JSGG20191129114029286; JSGG20201103153807021), the Talent Development Starting Fund from Shenzhen Government (HA11409030)."

4. Thank you for stating the following in theFinancial Disclosure section: 

"This work is partially supported by Shenzhen Science and Technology Research Fund (JCYJ20190806142203709; JSGG20191129114029286; JSGG20201103153807021), the Talent Development Starting Fund from Shenzhen Government (HA11409030)."

We note that one or more of the authors are employed by a commercial company: Chengdu Information Technology of Chinese Academy of Sciences Co Ltd 

Reviewers' comments:

Reviewer's Responses to Questions

**Comments to the Author**

1. Is the manuscript technically sound, and do the data support the conclusions?

Reviewer #1: Yes

Reviewer #2: Yes

2. Has the statistical analysis been performed appropriately and rigorously? 

Reviewer #1: Yes

Reviewer #2: No

3. Have the authors made all data underlying the findings in their manuscript fully available?

Reviewer #1: Yes

Reviewer #2: Yes

4. Is the manuscript presented in an intelligible fashion and written in standard English?

Reviewer #1: Yes

Reviewer #2: Yes

5. Review Comments to the Author

Reviewer #1: In the results required comparison table for exiting terms and proposed idea with all the parameters.

The Abstract is clear and implementation, conclusion meet the requirement.

The manuscript has been written in standard English.

Reviewer #2: The authors have developed drug activity and toxicity prediction based on Siamese networks and graph convolution to produce multitasking output for understudied targets to overcome the scarcity of drug data. The problem solved in the article is significant and relevant to the venue. The scientific novelty and contributions of this paper are good enough for publication. However, I have a few suggestions worth including to represent the article better.

Positive aspects of the paper:

This paper is well motivated and easy to follow.

Experiments are well designed, and the proposed algorithm works well compared to baseline algorithms.

Suggestion/comments:

The authors should highlight the novelty and motivation of their proposed method.

Further, they should highlight the challenges or limitations of the proposed work.

I suggest adding some statistical analysis tests to validate the results.

The paper needs proofreading to eliminate any typos errors, for example, on page 2, line 21.

6. PLOS authors have the option to publish the peer review history of their article (what does this mean?). If published, this will include your full peer review and any attached files.

Reviewer #1: **Yes: **Anandhan K

Reviewer #2: No

---

## [Author Response · Author response to Decision Letter 0]

3 Mar 2022

Dear academic editor and reviewers,

 I am very appreciated for your comments and suggestions. I have revised this manuscript and especially paid much attention to your comments and suggestions. The details of the revision are as follows:

 Replies to the academic editor:

 1. The style of the manuscript has been modified in the latex file to comply with the requirements of the template.

 2. The completeness and style of the reference list have been reviewed, and the retracted articles are not cited.

 3. The statement “There was no additional external funding received for this study.” has been included in our updated Funding Statement.

 4.“The funder provided support in the form of salaries for authors [XS, JZ, BC, HY and HX], but did not have any additional role in the study design, data collection and analysis, decision to publish, or preparation of the manuscript. The specific roles of these authors are articulated in the ‘author contributions’ section.” has been included in our updated Funding Statement.

5. "This does not alter our adherence to PLOS ONE policies on sharing data and materials.” has been included within our Competing Interests Statement.

 6. An updated Funding Statement and Competing Interests Statement have been included in our cover letter.

 Replies to Reviewer #1:

 1. The comparison tables (Tables 1 and 2) for exiting terms and proposed idea with all the parameters has been added in the results. 

 Replies to Reviewer #2:

 1. The novelty and motivation of the proposed method have been highlighted in the abstract as: Owing to an overall insufficiency and imbalance of drug data, it is hard to accurately predict drug activity and toxicity of multiple tasks by the existing models. To solve this problem, this paper proposed a two-stage transfer learning workflow to develop a novel prediction model, which can accurately predict drug activity and toxicity of the targets with insufficient observations.

 2. The challenges and limitations of the proposed work have been highlighted in the conclusion as: In order to make the model proposed in this paper can be used in common situation, it is essential to establish massive multi-task datasets for pretraining, which is also an interesting attempt for our future research work.

 3. Some statistical analysis tests, as illustrated in Figures 9, 10 and 11, have been added to validate the results.

 4. The manuscript has been proofread to eliminate typos errors. For example, line 21 on page 2 has been revised to read: For example, due to the difficulty of obtaining tumor tissue fragments of bone metastasis, the scarcity of sarcomatous-type tumors, and the difficulty in culturing bone tissue into cell lines, the number of cell models is insufficient.

We thank you in advance for considering our manuscript for publication in PLOS ONE.

Sincerely,

Xiaofei Sun

---

## [Decision Letter · Decision Letter 1]

15 Mar 2022

A feature transferring workflow between data-poor compounds in various tasks

PONE-D-22-00703R1

Dear Dr. Chen,

We’re pleased to inform you that your manuscript has been judged scientifically suitable for publication and will be formally accepted for publication once it meets all outstanding technical requirements.

Kind regards,

Sathishkumar V E

Academic Editor

PLOS ONE

Additional Editor Comments (optional):

Reviewers' comments:

Reviewer's Responses to Questions

**Comments to the Author**

1. If the authors have adequately addressed your comments raised in a previous round of review and you feel that this manuscript is now acceptable for publication, you may indicate that here to bypass the “Comments to the Author” section, enter your conflict of interest statement in the “Confidential to Editor” section, and submit your "Accept" recommendation.

Reviewer #1: All comments have been addressed

2. Is the manuscript technically sound, and do the data support the conclusions?

Reviewer #1: Yes

3. Has the statistical analysis been performed appropriately and rigorously? 

Reviewer #1: Yes

4. Have the authors made all data underlying the findings in their manuscript fully available?

Reviewer #1: Yes

5. Is the manuscript presented in an intelligible fashion and written in standard English?

Reviewer #1: Yes

6. Review Comments to the Author

Reviewer #1: All the correction has been updated. The reviewer comments are updated in the new version article.

7. PLOS authors have the option to publish the peer review history of their article (what does this mean?). If published, this will include your full peer review and any attached files.

Reviewer #1: **Yes: **ANANDHAN K

---

## [Editor Report · Acceptance letter]

21 Mar 2022

PONE-D-22-00703R1 

A feature transferring workflow between data-poor compounds in various tasks 

Dear Dr. Chen:

I'm pleased to inform you that your manuscript has been deemed suitable for publication in PLOS ONE. Congratulations! Your manuscript is now with our production department. 

Kind regards, 

on behalf of

Dr. Sathishkumar V E 

Academic Editor

PLOS ONE